# Expression of Huntingtin and TDP-43 Derivatives in Fission Yeast Can Cause Both Beneficial and Toxic Effects

**DOI:** 10.3390/ijms23073950

**Published:** 2022-04-01

**Authors:** Luis Marte, Susanna Boronat, Rubén Barrios, Anna Barcons-Simon, Benedetta Bolognesi, Margarita Cabrera, José Ayté, Elena Hidalgo

**Affiliations:** 1Oxidative Stress and Cell Cycle Group, Universitat Pompeu Fabra, C/Doctor Aiguader 88, 08003 Barcelona, Spain; luis.marte@vhir.org (L.M.); susanna.boronat@upf.edu (S.B.); ruben.barrios@upf.edu (R.B.); anna.barcons@para.vetmed.uni-muenchen.de (A.B.-S.); margarita.cabrera@urjc.es (M.C.); jose.ayte@upf.edu (J.A.); 2Institute of Bioengineering of Catalonia (IBEC), Baldiri Reixac 10–12, 08028 Barcelona, Spain; bbolognesi@ibecbarcelona.eu

**Keywords:** huntingtin, TDP-43, protein aggregation, fission yeast, neurodegenerative diseases

## Abstract

Many neurodegenerative disorders display protein aggregation as a hallmark, Huntingtin and TDP-43 aggregates being characteristic of Huntington disease and amyotrophic lateral sclerosis, respectively. However, whether these aggregates cause the diseases, are secondary by-products, or even have protective effects, is a matter of debate. Mutations in both human proteins can modulate the structure, number and type of aggregates, as well as their toxicity. To study the role of protein aggregates in cellular fitness, we have expressed in a highly tractable unicellular model different variants of Huntingtin and TDP-43. They each display specific patterns of aggregation and toxicity, even though in both cases proteins have to be very highly expressed to affect cell fitness. The aggregation properties of Huntingtin, but not of TDP-43, are affected by chaperones such as Hsp104 and the Hsp40 couple Mas5, suggesting that the TDP-43, but not Huntingtin, derivatives have intrinsic aggregation propensity. Importantly, expression of the aggregating form of Huntingtin causes a significant extension of fission yeast lifespan, probably as a consequence of kidnapping chaperones required for maintaining stress responses off. Our study demonstrates that in general these prion-like proteins do not cause toxicity under normal conditions, and in fact they can protect cells through indirect mechanisms which up-regulate cellular defense pathways.

## 1. Introduction

Even though protein aggregation is a reversible process in vivo (for a review, see [1]), proteostasis can be overwhelmed due to mutations or stress, and intra and extracellular protein inclusions are a hallmark of some pathological conditions and of aged cells. Thus, a common feature of many neurodegenerative disorders is the impairment of proteostasis, resulting in the amyloid aggregation of disease-specific proteins. The cause of toxicity and neurodegeneration may arise from a toxic gain of-function or loss-of-function of the disease protein. Genetic mutations can cause a protein to aggregate, e.g., the misfolded Huntingtin protein in Huntington’s disease, which suffers expansion of a poly-glutamine (polyQ) region. However, most cases of Parkinson’s, Alzheimer’s and amyotrophic lateral sclerosis (ALS) cannot be associated with any known mutations, so that specific proteins appear as aggregates in the absence of genetic changes. An environmental insult, such as exposure to toxic chemicals, may be the trigger of protein aggregation. The highest risk factor for most neurodegenerative diseases is aging since the capacity of the protein quality control (PQC) declines during this process.

Another important issue in the aggregation of these human proteins is the structure and location of the specific foci The general belief is that smaller, more soluble, oligomeric aggregates may be the most toxic species [2]. In recent years, several predictive algorithms have been developed to identify proteins with prion- or amyloid-like domains, as well as to predict their aggregation propensity; these computational approaches are often combined with experimental validations [3,4,5,6,7,8].

Regarding the molecular mechanisms that explain toxicity due to these aggregates, they are poorly understood. A first hypothesis is that these aggregates, and specifically the more heterogeneous amorphous oligomers, can display solvent-exposed hydrophobic surfaces, and these can cause aberrant interactions with other unrelated proteins; this sequestration of essential proteins could impair cell fitness [9,10,11]. A second possibility, which may not be mutually exclusive from the former but rather occur simultaneously, is that aggregating proteins may interfere with the regular components of the PQC system, so that chaperones or degrading activities [such as components of the ubiquitin-proteasome system (UPS)] may also be sequestered and leave unattended their natural substrates [12,13]. Again, both toxicity mechanisms may act in parallel.

Expansion of polyQ domains in specific proteins is the origin of several neurodegenerative diseases which are inherited as a dominant trait, with their onset and severity closely correlating with the length of the polyQ expansion [14]. In Huntington disease, the aggregation of a mutated protein, named Huntingtin (Htt) depends on the number of Q in the polyQ expansion [15,16,17]. Although the presence of aggregates often correlates with toxicity [18], it has been described that the oligomeric intermediates, formed before the constitution of the mature Htt deposits, are highly toxic and responsible for the cellular alterations observed in Huntington disease [19,20]. The molecular mechanisms leading to Huntington disease pathogenesis are not completely understood, and numerous factors may be involved.

Tar/DNA binding protein-43 (TDP-43) is an RNA-binding protein, with physiological nuclear localization, which accumulates in cytoplasmic inclusions in patients of ALS and other disorders. In fact, the pathological hallmark of ALS is the presence of protein inclusions in the patients’ motor neurons, containing superoxide dismutase, TDP-43 and Fused in Sarcoma (FUS). Most ALS patients do not display changes in the TDP-43-coding gene, even though some mutations are also found in a small number of ALS patients [21,22].

Most of these neurodegenerative disease-causing proteins do not have yeast orthologs. Nevertheless, the ectopic expression, or humanization, of these prion-like proteins in *Saccharomyces cerevisiae* has proven to be a useful strategy to study the aggregation and toxicity of these human proteins (for reviews, see [23,24]).

In particular, the ectopic expression of Htt.nQ and TDP-43 variants in budding yeast has been claimed to be efficient to study the molecular bases of protein aggregation and of cellular toxicity. To provide insights into the mechanisms causing Htt protein aggregation, the laboratory of Susan Lindquist proposed in 2000 the use of a fluorescent chimera, Htt.nQ-GFP, to follow protein aggregation in yeast [25]. Since then, many laboratories have used this heterologous system to study both aggregation propensity as well as toxicity [26,27,28,29,30,31]. Similarly, TDP-43 has also been expressed in budding yeast [32,33]. In both cases, the isolation of deletion or over-expression suppressors of aggregation or toxicity has been described extensively, as well as the effect of small drugs on blocking either aggregate formation or cellular defects. Another classical strategy to understand aggregate formation and/or toxicity has been to perform quantitative mutagenesis to score aggregation propensity (and/or derived toxicity) of human disease proteins. This has been recently performed expressing a collection of mutagenized TDP-43 prion domain library in budding yeast [34].

*Schizosaccharomyces pombe* has not been extensively used as a model system to study PQC. Using misfolding reporters, we have recently described the formation and fate of protein aggregate centers, which appear after heat shock [35]. Thus, non-irreversibly misfolded proteins, with the aid of the Hsp40/Hsp70 chaperones Mas5/Ssa2, are sequestered upon heat shock into discrete protein aggregate-like centers (PACs) to escape from degradation; only when the low temperature is recovered the Hsp104 disaggregase refolds the proteins from PACs to their soluble forms [35]. In fact, the Mas5/Ssa2 couple seems to be the master regulator of PQC in *S. pombe*: it is essential for the maintenance and folding of intrinsically unstable proteins in the absence of stress [36], and it also maintains the stress regulators Hsf1 [37] and Sty1 (a MAP kinase regulating the common environmental stress response) [38] inactive unless heat shock is applied [36,39].

Regarding the ‘humanization’ of *S. pombe* to study the bases of neurodegenerative diseases, Supattapone and colleagues expressed Htt in fission yeast, to conclude that Htt.103Q can aggregate but cannot exert toxicity in this unicellular eukaryote [40,41]. The distribution of polyQ-containing proteins varies from one cell type to another, being surprisingly small in the fission yeast proteome (only 0.07% of all proteins in *S. pombe* display long polyQ stretches). The fact that expression of Htt with long (but not short) polyQ tails in fission yeast produces intracellular aggregates but not toxicity would be consistent with the idea that the bases of the toxicity exerted by protein aggregation is the aberrant sequestration of endogenous and essential prone-to-aggregate proteins. Whether other non-polyQ-containing human proteins, such as TDP-43, cause or not toxicity in *S. pombe* has not yet been studied.

Here, we express not only Htt variants but also TDP-43 derivatives in fission yeast. In both cases we have detected toxicity caused by aggregation, but only when the proteins are expressed at extremely high levels. Some TDP-43 derivatives, more soluble and oligomeric than others, can also elicit more damaging effects in fission yeast than Htt variants, suggesting that the physical state and type of aggregates are critical for damage. When toxicity occurs, the protein degradation machineries, namely UPS and autophagy, were not affected. Deletion of the genes coding for the chaperones Mas5 or Hsp104 had opposite effects on Htt aggregates, while TDP-43 aggregation tendency resulted unaffected. Importantly, we demonstrate that moderate expression of the aggregating Htt.103Q causes a beneficial impact on fission yeast lifespan, through sequestration of the Hsp40 chaperone Mas5 and consequent activation of a Pyp1 and Sty1-dependent stress response cascade.

## 2. Results

### 2.1. Expression of Htt.Qn-GFP Versions in Fission Yeast

We expressed in *S. pombe* the Htt chimeras described by the Lindquist lab, which include the N-terminal domain of Htt fused to 25, 47, 97 and 103 polyQ repeats, followed by a proline-rich domain of Htt and the GFP tag [25] (Figure 1A). These proteins were expressed from constitutive or inducible promoters of different strengths (Figure 1A).

For constitutive expression, the Htt^NTD^.nQ-GFP-coding genes were integrated into the *S. pombe* genome and expressed under the control of the constitutive *tpx1* and *sty1* promoters. Based on Western blot analysis using the internal endogenous protein Pap1, with a constitutive concentration of 45 nM [42], and specific polyclonal antibodies (Appendix A), we determined that the intracellular protein concentrations of the chimeras from *tpx1* promoter is in the order of 3 µM (Figure 1B and Table 1), while the protein concentrations of the Htt chimeras expressed from the *sty1* promoter were in the order of 1 μM (Table 1). Note that a faint slower migrating band was observed for all Htt derivatives, which we suspect could be arising from post-translational modifications or structural differences affecting electrophoretic mobility.

We analyzed by fluorescence microscopy the aggregation profiles of the different Htt^NTD^.nQ-GFP fusion proteins. The 25Q and 47Q chimeras displayed diffuse cytosolic distribution when expressed either from the *tpx1* or the *sty1* promoters. On the contrary, Htt^NTD^.103Q-GFP expressed from the strong *tpx1* promoter aggregated into discrete foci (Figure 1C). 60% of cells growing in minimal medium and expressing *tpx1* promoter-driven Htt^NTD^.103Q-GFP contained aggregates (only 20% if cells were growing in rich medium).

In order to determine whether the constitutive expression of Htt^NTD^.nQ-GFP could affect cell fitness, we spotted cells expressing Htt^NTD^.nQ-GFP, and we did not observe any significant growth defect compared to the wild-type strain, either in the absence or presence of environmental stressors (Appendix A). Moreover, it has been described that depletion of the proline rich region of Htt is required to induce toxicity in budding yeast [27,44]. However, we expressed Htt chimeras lacking this region and did not observe any deleterious effect on cell growth (Appendix A).

We next studied the expression of Htt^NTD^.nQ-GFP under the control of the inducible *nmt1* (no message in thiamine) promoter, which triggers the expression of downstream genes in the absence of thiamine. In order to achieve different protein levels, we expressed the constructs from integrative and episomal plasmids (7–8 plasmid copies per cell [43]). As shown in Figure 1D, protein levels of the chimeras from *nmt1*-driven episomal plasmids reached the highest levels (around 9 μM), about three times higher than the *tpx1* constitutive expression (3 μM), whereas the integrative conditional expression system allowed an intermediate level (4 μM) (Figure 1D and Table 1). Next, we visualized by fluorescence microscopy the aggregation patterns of these *nmt1*-driven Htt^NTD^.nQ-GFP chimeras. As previously shown with the constitutive *tpx1* promoter, expression of Htt^NTD^.103Q-GFP (but not 25Q or 47Q) caused the formation of aggregation foci in 30% of the cells. Only when the highest expression levels, ~9 μM, were reached using episomal *nmt1*-driven plasmids, then fluorescent foci could be detected in the 25Q, 47Q and 103Q-expressing cells (Figure 1E). The foci created by the 25Q and 47Q chimeras had a shape, compactness and brightness different from the 103Q ones, and the number of cells with aggregates also varied depending on the fusion protein (Figure 1E). Regarding the Htt^NTD^.103Q-GFP chimera, it formed aggregates when expressed at 3, 4 and 9 μM, but the percentage of cells with aggregates was inversely proportional to the intracellular concentration, even though the size of aggregates was larger with higher concentrations (compare Figure 1C,E).

Next we investigated the effects of the inducible expression of the chimeras on cell growth. As shown in Figure 1F, proteins expressed from the integrative plasmids did not have any impact on fission yeast growth on solid plates, while the high concentrations of the chimeras when expressed from episomal plasmids jeopardized cell growth to different extents. In liquid cultures, expression of the soluble GFP moiety alone from episomal plasmids was sufficient to induce some growth defects (Figure 1G), even though the protein does not accumulate into foci (Appendix A). The three polyQ Htt chimeras displayed further capacity to impair cell growth, 47Q having the strongest toxic effects and 25Q the weakest (Figure 1G).

In conclusion, aggregate formation of the Htt^NTD^.nQ-GFP chimeras depends on both the concentration and the polyQ length. In addition, the expression of Htt^NTD^.nQ-GFP can lead to toxicity but only if they are expressed at extremely high levels, around 9 µM.

### 2.2. Variants of TDP-43 Cause More Toxicity in Fission Yeast than Htt^NTD^.103Q

We generated a series of constructs to express human TDP-43 in its wild-type version and with the mutation M322K in the C-terminal prion-like domain (PRD), fused to a C-terminal yellow fluorescent protein (YFP) (Figure 2A). This mutation affects the number and location of the aggregation foci, decreases TDP-43 hydrophobicity and increases TDP-43 toxicity in budding yeast [34]. We expressed them under the control of the inducible *nmt1* and *nmt41* promoters, using again integrative and episomal plasmids. The *nmt41* promoter is 10-fold weaker than *nmt1* [45]. As expected, the protein levels of the TDP-43-YFP variants and the Htt^NTD^.nQ-GFP chimeras were similar when using the same expression system (*nmt1* promoter and episomal plasmids) (Figure 2B and Appendix A). The steady-state concentrations reached with wild-type and mutant TDP-43 are indicated in Table 1.

We studied the sub-cellular distribution and aggregation propensity of TDP-43 in *S. pombe* by fluorescence microscopy. Regarding wild-type TDP-43-YFP, 100% of the cells expressing it from the integrative plasmid displayed nuclear aggregates, but when TDP-43-YFP was expressed at higher concentrations (*nmt1*) from episomal plasmids the aggregates were also present in the cytosol. On the other hand, cells expressing mutant TDP-43.M322K-YFP displayed nuclear and cytosolic foci in both the integrative and the episomal versions (Figure 2C). When both forms of TDP-43 were expressed from integrative plasmids with the weaker *nmt41* promoter, only the wild-type form collapsed into foci (Figure 2C); probably the M322K variant did not aggregate due to its increased solubility [34].

We next tested whether expression of TDP-43 causes toxicity in *S. pombe*. Similar to Htt^NTD^.103Q-GFP, wild-type TDP-43-YFP only affected growth on solid media when expressed at the highest levels (from episomal plasmids and the strong *nmt1* promoter) (Figure 2D). However, mutant TDP-43.M322K-YFP caused toxicity also at lower levels (Figure 2D, *nmt1* integrative). Growth of liquid cultures were more affected by the expression of the TDP-43 derivatives; only the lower levels of expression accomplished with integrative plasmids and the weaker *nmt41* promoter did not affect growth rates of fission yeast (Figure 2E). This liquid growth assay demonstrated again that TDP-43.M322K was more toxic than wild-type TDP-43 at all expression levels (Figure 2E). Therefore, we suggest that the presence of cytosolic inclusions of TDP-43.M322K, or the different physicochemical properties of the aggregates (i.e., lower hydrophobicity) are linked to toxicity. We also conclude that the mutant form TDP-43.M322K, as demonstrated in *S. cerevisiae* [34], causes more toxicity than the wild-type construct.

As mentioned in the Introduction, the toxicity caused by the expression of Htt and other aggregation-prone proteins in *S. cerevisiae* is a rather controversial issue, with some laboratories claiming that not even Htt^NTD^.103Q-GFP can affect cell fitness [25]. To shed light into this issue and to determine whether budding and fission yeast are differentially affected by the expression of TDP-43, we have compared the effect of TDP-43 expression in *S. pombe* and *S.*
*cerevisiae*. First, we expressed the same two constructs under the control of the galactose-inducible *GAL1* promoter in budding yeast. We confirmed that the growth of *S. cerevisiae* is impaired by expression of wild-type and mutant TDP-43, the later causing more dramatic effects (Figure 2F). The wild-type TDP-43-YFP protein levels reached in budding yeast using the *GAL1* promoter are similar to those accomplished in fission yeast from the *nmt1* promoter using integrative plasmids (Figure 2G); those levels barely triggered toxicity in fission yeast (Figure 2D). These results suggest that the expression of TDP-43 can cause toxicity in both yeast types and the M322K mutation significantly enhances this toxic effect. However, at the same protein levels TDP-43 exerts more toxicity in *S. cerevisiae* than in *S. pombe*, supporting the claim that *S. pombe* has a more “sheltered” proteome than *S. cerevisiae,* at least regarding the toxicity caused by expression of Htt.103Q [40,41] and, now, TDP-43. Alternatively, the PQC system might be more robust in fission yeast than in budding yeast.

### 2.3. The Intracellular Steady-State Levels of Htt^NTD^.103Q-GFP and TDP-43-YFP Derivatives Are Not Dependent on the UPS or Autophagy

In human cells, expression of an aggregation-prone protein can cause toxicity due to (i) the inactivation of the otherwise soluble protein, (ii) the sequestration and inactivation into the aggregates of other essential proteins; or (iii) the sequestration of PQC components which, by joining the aggregates foci, disregard their housekeeping functions. Regarding the latter, it has been reported that aggregate deposits can impair the normal activity of some PQC components, affecting the proper proteostasis of cells. In particular, attempts of degradation of the aggregates by the UPS or autophagy could jeopardize the natural activities of these systems.

First, we tested whether our reporters have a turnover similar to other proteins controlled by the UPS. We blocked protein synthesis by treating cells expressing Htt^NTD^.nQ-GFP under the control of the constitutive *tpx1* promoter with the translation inhibitor cycloheximide. The Htt derivatives seem to be extremely stable, since 14 h of translation inhibition did not affect the steady-state proteins levels (Figure 3A). The same occurred with the TDP-43 derivatives (Figure 3B). The high stability of our reporters suggests that the UPS is not actively degrading them, and therefore its function may not be compromised.

We expressed our Htt derivatives in mutants affected in the UPS, such as some carrying temperature-sensitive alleles of essential proteasome components (Appendix A). The Htt derivatives behaved as in the wild-type background, with no significant changes in the number of cells with aggregation foci (Appendix A), or in the protein levels as determined by Western blot (data not shown). Similarly, addition of the proteasome inhibitor bortezomib did not affect the steady-state levels of the Htt chimeras (data not shown). We conclude that the UPS is not involved in the degradation of this aggregation-prone reporters, and probably this PQC function is unaffected by their expression.

We then tested if autophagy is involved in the elimination of the Htt^NTD^.nQ-GFP or TDP-43-YFP chimeras. Extracts from wild-type and Δ*atg1* strains (autophagy is abolished in this mutant [46]) displayed very similar levels of the proteins (Appendix A), and toxicity is not significantly exacerbated in this strain background (Appendix A), which indicates that autophagy inhibition has no effect on the steady-state levels of Htt^NTD^.nQ-GFP or TDP-43-YFP.

Autophagy is strongly up-regulated upon nitrogen starvation, and Pgk1-GFP vacuolar cleavage has been recently optimized as a reporter of bulk autophagy in fission yeast [47] (Figure 3C). While some cleavage of both Htt^NTD^.103Q-GFP and Htt^NTD^.47Q-GFP chimeras was observed upon nitrogen starvation with the generation of the GFP moiety, the amount of GFP released is similar for the aggregating 103Q than for the soluble 47Q, suggesting that 103Q aggregates are not specifically targeted to the vacuole through autophagy. In conclusion, it is unlikely that other biological roles of autophagy may be jeopardized by the expression of the aggregation-prone Htt or TDP-43 derivatives.

### 2.4. Several PQC Components Modulate the Aggregation Propensity of Htt^NTD^.103Q-GFP but Not of TDP-43-YFP Derivatives

Since the first report by the Lindquist lab describing ‘humanized’ budding yeast expressing Htt derivatives [25], the effect of different chaperones and PQC components on Htt^NTD^.nQ-GFP aggregation and/or toxicity has been extensively studied. As described in the Introduction, we have recently characterized the role of the Hsp40/70 couple Mas5/Ssa2 in the folding of non-native intermediates during normal conditions [36] and on the assembly of PACs during heat stress to avoid misfolding protein degradation [35]. We tested whether deletion of the *mas5* gene would have an impact on the aggregation of our reporters in fission yeast.

We expressed two Htt^NTD^.nQ-GFP derivatives in strain Δ*mas5*. As shown in Figure 4A, the lack of Mas5 significantly exacerbated the number of cells with aggregates of the 103Q chimera, and induced foci formation of the 47Q protein, which did not coalescence into aggregates in a wild-type background (Figure 4A). Furthermore, we detected a significant proportion of ∆*mas5* cells with aggregates when Htt^NTD^.103Q-GFP was expressed from the weak *sty1* promoter (Figure 4A). We conclude that the Hsp40 Mas5 contribute to the folding of the Htt derivatives, so that in the absence of Mas5 both 47Q and 103Q proteins display enhanced tendency to misfold and aggregate.

As explained in the Introduction, Hsp104 is required in fission yeast for the disaggregation of PACs after heat shock recovery [35]. As described before in Htt-humanized budding yeast [25], the over-expression or the lack of Hsp104 block aggregate formation of Htt^NTD^.103Q-GFP also in fission yeast (Figure 4B,C). This suggests that the Hsp104 chaperone is not only involved in the disaggregation of protein aggregates, but it also contributes to aggregate formation [35]. We further tested the effect of several deletion mutants coding for other PQC components (such as *hsp16*, *hul5*, *ubp8*, *san1*, *ubp3*, *bag101*, *rpt4*, *def1*, *SPBC17A3.05C*, *ssa1*, *ubp16*, *ubr1* or *ltn1*) on the aggregation profile of Htt^NTD^.nQ-GFP derivatives; none of them had a significant effect on the number or profile of Htt aggregation (Appendix A).

We next evaluated the impact of these important chaperones, Mas5 and Hsp104, in TDP-43 aggregate formation. As shown in Figure 4D, neither the absence of Mas5 nor of Hsp104 affected the aggregation capacity of TDP-43-YFP derivatives; 100% of the cells lacking Mas5 or Hsp104 displayed TDP-43 foci. These findings strengthen the notion that Htt and TDP-43 propensity to form aggregates is very different in nature, Htt derivatives being more similar to the intrinsically unstable proteins which only form aggregates upon heat shock with the assistance of chaperones. The capacity of TDP-43 variants to form aggregates seems to be intrinsically strong so that it can dispense the presence of chaperones.

In line with this hypothesis, we compared the solubility of Htt and TDP-43 derivatives in whole protein extracts of cells expressing the proteins to similar levels. Both TDP-43 and TDP-43.M322K were highly enriched in the Tween insoluble fraction, whereas significantly lower levels of Htt^NTD^.47Q-GFP and Htt^NTD^.103Q-GFP were found in the insoluble/pellet fraction (P in Figure 4E). We propose that TDP-43 derivatives have strong and intrinsic tendency to aggregate, while a fraction of the intracellular Htt derivatives, with weaker aggregation propensity, may be protected from aggregation by the fission yeast PQC system.

### 2.5. Expression of Non-Toxic Concentrations of Htt^NTD^.103Q-GFP Extends Lifespan

Htt and TDP-43 derivatives can exert toxicity to fission yeast but only when expressed at extremely high doses (Figure 1 and Figure 2). In an attempt to find other stress-dependent phenotypes, either toxic or beneficial, linked to the expression of these aggregation-prone proteins, we measured chronological lifespan of cells expressing some of these chimeras. The viability to survive in the stationary phase has been described to depend on glucose deprivation pathways and on stress cascades [48,49]. As controls, we used wild-type cells and strain Δ*pyp1*, which lacks the main phosphatase of the Sty1 kinase, displays constitutive activation of the Sty1-dependent stress program [50], and has been characterized as a long-lived mutant [51]. Surprisingly, expression of *tpx1*-driven Htt^NTD^.103Q-GFP, but not Htt^NTD^.47Q-GFP, had a beneficial effect on chronological aging (Figure 5A). Expression of the 103Q chimera in cells lacking Hsp104, where this protein did not coalescence into aggregation foci (Figure 4C), did not promote enhanced viability during stationary phase (Figure 5B). Death at the stationary phase may be dependent on oxidative damage, so that the levels of carbonylated proteins were elevated in stationary phase cultures of wild-type or 47Q-expressing cells; on the contrary, the levels of proteins carbonyls in cells expressing 103Q were significantly lower (Figure 5C). TDP-43 derivatives expressed to higher levels as 103Q using the *tpx1* promoter (Appendix A) formed aggregation foci (Appendix A) but did not contribute to longevity (Figure 5D), confirming the different nature of the Htt and TDP-43 aggregates.

We have recently established that the Mas5/Ssa2 chaperones are required for the maintenance and folding of intrinsically unstable proteins such as Pyp1, a phosphatase which keeps the MAP kinase Sty1 inactive in the absence of stress [36,52]. It has been previously described that the over-expression of aggregating Htt in *S. cerevisiae* can exert toxicity by sequestrating the Hsp40 chaperone Sis1 [12]. We tested whether expression of Htt^NTD^.103Q-GFP in *S. pombe* would sequester Mas5 away from its role in Pyp1 folding, so that the Sty1-dependent stress program would be engaged promoting an elongated lifespan. We have four evidences suggesting that this is the case. First, stress-dependent activation of Sty1 was stronger at short times when cells are expressing the 103Q derivative (compare 5 min time point in Figure 5E). Second, the basal levels of expression of *hsp9*, a stress gene which are up-regulated by Sty1 in a stress-dependent manner, was significantly higher in cells expressing Htt^NTD^.103Q-GFP but not Htt^NTD^.47Q-GFP, as well as the H_2_O_2_-induced levels of other stress genes such as *ctt1*, *gpx1* and *gpd1* (Figure 5F). Third, deletion of *atf1*, coding for the transcription factor up-regulating the stress genes, suppressed the beneficial effect of Htt^NTD^.103Q-GFP on longevity (Appendix A). And forth, expression of Htt^NTD^.103Q-GFP in cells carrying an integrated extra copy of the *pyp1* gene and therefore expressing twice the amount of *pyp1* mRNA (Appendix A) did not confer any advantage in chronological aging and displayed wild-type lifespan (Figure 5G). These experiments suggest that expression of Htt^NTD^.103Q-GFP can be partially sequestering the Mas5 chaperone, without leading to any basal toxicity hallmark but rather inactivating some Pyp1 molecules and up-regulating the anti-aging Sty1 response.

## 3. Discussion

We have expressed Htt^NTD^.nQ-GFP and TDP-43-YFP chimeras in fission yeast to different levels, and studied their propensity to form aggregates and the phenotypic consequences of their expression. Our conclusion is that cell toxicity only occurs when these human proteins were expressed at extremely high concentrations. In general, TDP-43 derivatives displayed more propensity to aggregate and to affect cell fitness that Htt^NTD^.nQ-GFP variants. Surprisingly, the aggregating derivative Htt^NTD^.103Q-GFP was able to cause beneficial effects elongating the lifespan.

Regarding toxicity, we want to emphasize that the aggregation prone Htt^NTD^.103Q-GFP, TDP-43-YFP and TDP-43.M322K-YFP variants could induce toxicity, but only when expressed at extremely high concentrations. Htt^NTD^.103Q-GFP had to be expressed at 9 µM to cause growth defects, while TDP-43 derivatives conferred toxicity at lower concentrations (5 and 4 µM for wild-type and M322K forms, respectively). As a reference, the translation elongation factor EF1a (coded from *SPAC23A1.10*) is the most abundant protein during vegetative growth of the fission yeast proteome [42], more than a million copies per cell; considering that the cell volume of *S. pombe* is around 100 fl [53], this indicates that EF1a has an intracellular concentration of 19 μM during normal growth conditions. In other words, these human proteins, Htt^NTD^.103Q-GFP and TDP-43-YFP, are toxic when expressed in fission yeast at close to maximum concentrations. Likewise, our results indicate that the expression of wild-type TDP-43-YFP is more deleterious for cell fitness than the expression of the Htt^NTD^.nQ-GFP variants, since at the same levels, TDP-43 exerted more growth defects. The expression of TDP-43-YFP drives to aggregation in 100% of the cells, while expression of Htt^NTD^.103Q-GFP does not. Thereby, the two reporters display characteristic and differential features regarding both aggregation propensity and capacity to trigger damage.

As described in the Introduction, the group of Supattapone had previously used *S. pombe* to analyze the effect of the expression of the Htt^NTD^.nQ-GFP chimeras in cell fitness [40,41]. They proposed that, unlike budding yeast, aggregation of these polyQ proteins does not trigger toxicity in fission yeast. Since *S. pombe* has only three endogenous proteins with polyQ tracks greater than 10 and *S. cerevisiae* contains 79, they proposed that the cause of toxicity in budding yeast could be due to the sequestration of essential endogenous Q-rich proteins leading to their inactivation. However, a recent report by the Cowen lab using *Candida albicans* as a model system demonstrated that expression of Htt derivatives with as many as 230 polyQ did not trigger toxicity in this yeast, even though the Q-rich proteome of *C. albicans* is very similar to that of *S. cerevisiae* [54]. A proper comparison of the intracellular concentrations at which each Htt derivative is expressed in each model system, as we have done here for TDP-43, would be highly recommended for these two studies in *S. cerevisiae* and in *C. albicans*.

Only three proteins in this study displayed aggregation foci at different concentrations: Htt^NTD^.103Q-GFP, and the two TDP-43 derivatives. Their aggregation properties differed in many aspects. First, only a small proportion of cells expressing Htt^NTD^.103Q-GFP displayed aggregation foci, while 100% of the cells were decorated with TDP-43-derived foci. Second, in general TDP-43 foci were brighter, and the number of foci per cell was larger than for Htt^NTD^.103Q-GFP expressing cells. Third, while Mas5 and Hsp104 contributed to Htt^NTD^.103Q-GFP aggregation (Mas5 decreasing foci formation and Hsp104 contributing to foci formation), these chaperones did not have any effect on TDP-43 aggregation propensity. And forth, biochemical fractionation of insoluble proteins from cell extracts enriched more efficiently TDP-43 than Htt^NTD^.103Q-GFP aggregates. We propose that TDP-43 derivatives have stronger aggregation propensity than Htt derivatives, and newly synthesized TDP-43 polypeptides cannot be rescued/refolded by general PQC components such as Mas5, and they unavoidably fall into the aggregating fate.

During the course of this study, it was striking to find out that expression of Htt^NTD^.103Q-GFP was beneficial for lifespan. Based on the experiments presented here, we propose the model depicted in Figure 6. We have recently described that the Hsp40/70 chaperones Mas5/Ssa2 contribute to the correct folding of the phosphatase Pyp1, an intrinsically disordered protein, which maintains the Sty1 MAP kinase dephosphorylated and inactive (Figure 6A). Expression of moderate levels of Htt^NTD^.103Q-GFP may request the assistance of Mas5 for proper folding, kidnapping it from some of its regular clients. As a consequence, unattended Pyp1 will accumulate in a misfolded conformation which will be degraded by the UPS; Sty1 will be then more prone to become active and to trigger hundreds of stress genes in an Atf1-dependent manner, and this will contribute to an elongated lifespan (Figure 6B). TDP-43 or non-aggregating Htt^NTD^.nQ-GFP variants could not trigger this beneficial lifespan extension, probably because they do not sequester as many molecules of Mas5 as Htt^NTD^.103Q-GFP does. Interestingly, a connection between the expression of polyQ proteins in budding yeast and the up-regulation of the Hsf1-dependent anti-stress gene expression program has recently been proposed [55,56]. We also detected a mild up-regulation of some Hsf1-dependent genes by expression of Htt^NTD^.103Q-GFP (Appendix A), since Mas5 also participates in the sequestration/inactivation of this heat shock-sensing transcription factor [37]. Whether expression of aggregating proteins affect the PQC system to cause toxicity or beneficially exacerbate anti-stress gene expression programs will have to be studied for each particular biological system.

## 4. Materials and Methods

### 4.1. Growth Conditions, Yeast Strains and Plasmids

*S. pombe* was grown in rich medium (YE5S), synthetic minimal medium (MM) or MM plus thiamine (2 µM) as previously described [57]. *S. cerevisiae* was grown in synthetic complete media lacking uracil as described [58]. Origins and genotypes of strains and plasmids used in this study are outlined in Appendix A respectively. C-terminal tagging of genes was done by homologous recombination with PCR fragments from pFA6a plasmid derivatives [59]. Deletion mutant strains are from the Bioneer collection [60] and prototrophic or leu- derivatives were obtained by crossing the original auxotrophic deletion mutants with wild type strain 972 or with HM123. Strains expressing Htt^NTD^.nQ-GFP or TDP43-YFP were obtained by transformation of HM123 or deletion mutant strains with episomal or with linearized integrative plasmids and selection for Leu+ strains. *S. cerevisiae* strains have been used before [34]. Plasmids expressing Htt^NTD^.nQ-GFP were obtained by cloning Htt^NTD^.nQ-GFP from p426/PQn plasmids [25] kindly provided by E. Cabiscol into either episomal pREP plasmids (Maundrell, 1993) (p659.nQ.3x) or into empty integrative pREP plasmids (p688.nQ.3x’) or pREP-derived plasmids in which the *nmt* promoters has been substituted with the constitutive *sty1* [61] (p499.nQ) or *tpx1* [62] promoters (p503.nQ). Plasmids p521.nQΔP were obtained by cloning Htt^NTD^.nQΔP-GFP from p414GPD.nQ [44], kindly provided by A. Bertolotti, into integrative pREP derivatives carrying the *tpx1* promoter. Plasmids p723.nQ were generated by adding a cassette coding for a *kan::MX6* resistance into p503.nQ plasmids. *S. cerevisiae* plasmid expressing TDP-43.M322K-YFP was obtained by PCR mutagenesis of pRS416-TDP-43-YFP [63] as described before [34], and cloned into different *S. pombe* backbones, either empty episomal pREP plasmids (p660.3x and p660.41x), empty integrative pREP (p710.3x′ and p710.41x′) or pREP derivatives with *tpx1* promoter (p689). Plasmid p520.41x; was generated by amplifying *hsp104* coding sequence from genomic DNA and by cloning it into an empty pREP episomal plasmid with the *nmt41* promoter. Plasmid p690.3x was constructed by cloning GFP in a pREP episomal plasmid. Plasmid p819′ was constructed by cloning *pyp1* gene into an integrative pREP derived plasmid with the *nmt1* promoter replaced by the *pyp1* promoter and by inserting an *hph::MX6* cassette.

### 4.2. Native and TCA Extracts and Western Blot

*S. pombe* cells were grown in MM or YE5S to an OD_600_ of 0.5. *S. cerevisiae* cells were grown in synthetic complete media lacking uracil plus 2% glucose to an OD_600_ of 0.5 and then 2% of galactose was added for for 4–6 h. Modified trichloroacetic acid (TCA) and native extracts were prepared as previously described [47]. Samples were separated by SDS-PAGE and detected by immunoblotting. GFP-tagged and YFP-tagged proteins were visualized with a polyclonal (homemade) [64] or monoclonal anti-GFP (Takara); phosphorylated Sty1, with polyclonal anti-p38 (Cell Signaling Technology, Danvers, MA, USA); HA-tagged proteins are visualized with monoclonal antibody (12CA5; homemade), and both anti-Sty1 polyclonal antibody or Ponceau staining were used as loading controls.

### 4.3. Fluorescence Microscopy

*S. pombe* cells were grown in MM or YE5S to an OD_600_ of 0.5. Cells were harvested by centrifugation 1 min at 3000 rpm and visualized at room temperature as described before [35]. Briefly, images were acquired using a Nikon Eclipse 90i microscope equipped with differential interference contrast optics, a PLAN APO VC 100× 1.4 oil immersion objective, an ORCA-II-ERG camera (Hamamatsu, Herrsching am Ammersee, Germany), excitation and emission filters GFP-4050B and mCherry-C (Semrock, West Henrietta, NY, USA) and image image acquisition software Metamorph 7.8.13 (Gataca Systems, Massy, France). Processing of all images was performed using Fiji (ImageJ, National Institutes of Health, Bethesda, MD, USA) [65]. To determine the number of cells with aggregates, at least 50 cells were counted from acquired images. Appendix A compiles the results on % of cells with protein aggregates upon expression of the different protein chimeras in different backgrounds assayed.

### 4.4. Sensitivity Assays on Plates—Survival upon Protein Expression, Stress or Chronological Aging

To measure toxicity linked to protein expression in *S. pombe* strains expressing different versions of Htt^NTD^.nQ-GFP or TPD-43-YFP, the survival on solid plates was performed by serially diluting logarithmically growing cells in MM plus 2 µM thiamine and spotting them in MM plates with or without 20 µM thiamine as previously described [66]. For *S.cerevisiae,* strains were grown in synthetic complete media lacking uracil with 2% glucose to an OD_600_ of 0.5, serially diluted 1:10 six times from an initial concentration of 0.5 and spotted in synthetic complete media plates lacking uracil with 2% glucose or 2% galactose. In both cases, plates were incubated at 30 °C during 2 to 3 days.

For chronological aging survival, sensitivity plate assays were basically performed as described previously. Cells were grown in YE5S to logarithmic phase and 3–6 days after reaching stationary phase. At these points, cells were serially diluted and spotted on YE5S plates. Plates were imaged after 3–5 days of growth at 30 °C.

For stress survival, cells were grown to logarithmic phase in YE5S, pre-treated for 2 h at 37 °C and heat-shocked at 50 °C for 20 min. Cells were serially diluted and plated on YE5S solid plates. For stress resistance, cells were grown to logarithmic phase in YE5S, serially diluted and plated on YE5S plates with or without 1 mM H_2_O_2_ at left to grow at 30 °C or on YE5S plates and left to grow at 37 °C.

### 4.5. Sensitivity Assays in Liquid Cultures—Recording of Growth Curves

Yeast cells were grown in MM with 2 µM thiamine at 30 °C to an OD_600_ of 0.5 at which point thiamine was removed, cells were diluted to OD_600_ 0.1 and growth was recorded using an assay based on automatic measurements of optical densities, as previously described [66].

### 4.6. Protein Solubility Assay

Cells from 50 mL culture (OD_600_ 0.5) grown in MM were harvested by centrifugation and resuspended in 250 µL lysis buffer A [25 mM Tris-HCl pH 7.5, 0.1% Tween, 50 mM KCl, 10 mM MgCl_2_, 1 mM EDTA, Glicerol 5%, protease inhibitor cocktail (Roche, Basel, Switzerland), 1 mM PMSF and aprotinin (0.03–0.07 TIU)]. After addition of glass beads, cells were lysed using Vortex Genie 2 (Scientific Industries, Bohemia, NY, USA). Lysates were centrifuged at 2000× *g* for 5 min at 4 °C, and an aliquot of the resulting supernatant (Total, T) was then centrifuged at 16,000× *g* for 10 min to obtain pellet (P) and supernatant (S) fractions. Pellets were resuspended in a volume of 1xSDS loading buffer equal to the volume of the supernatant. Supernatants and pellets in SDS sample buffer were incubated for 10 min at 37 °C, loaded onto SDS-polyacrylamide gel electrophoresis (SDS-PAGE) gels, and subjected to immunoblot analysis.

### 4.7. Determination of Total Protein Carbonyls

Yeast cells growing logarithmically in YE5S and at day 1 and day 4 of stationary phase were collected and processed as described before [67].

### 4.8. Reverse-Transcription and Quantitative qPCR

Total RNA extraction was performed from logarithmically growing cells, treated or not with H_2_O_2_, in YE5S, as previously described [68]. DNase I treatment of RNA, cDNA synthesis and RTqPCR were performed as previously described [69].

### 4.9. Quantification and Statistical Analysis

Unless otherwise stated, all experiments were performed at least three times and representative experiments were shown. Data are presented as mean ± SD; * *p* < 0.05; ** *p* < 0.01; *** *p* < 0.001; **** *p* < 0.0001 (unpaired Student’s *t*-test). Graphs and statistical analysis were performed with Prism v 7.04 (GraphPad Software, San Diego, CA, USA). To determine the number of cells containing Htt^NTD^.nQ-GFP or TDP43-YFP aggregates at least 50 cells were counted in each condition from several acquired images.

### 4.10. Data Availability

All images included in the main and Appendix A figures are available as Mendeley dataset (doi:10.17632/34mxxhv9mg.1).

## Figures and Tables

**Figure 1 ijms-23-03950-f001:**
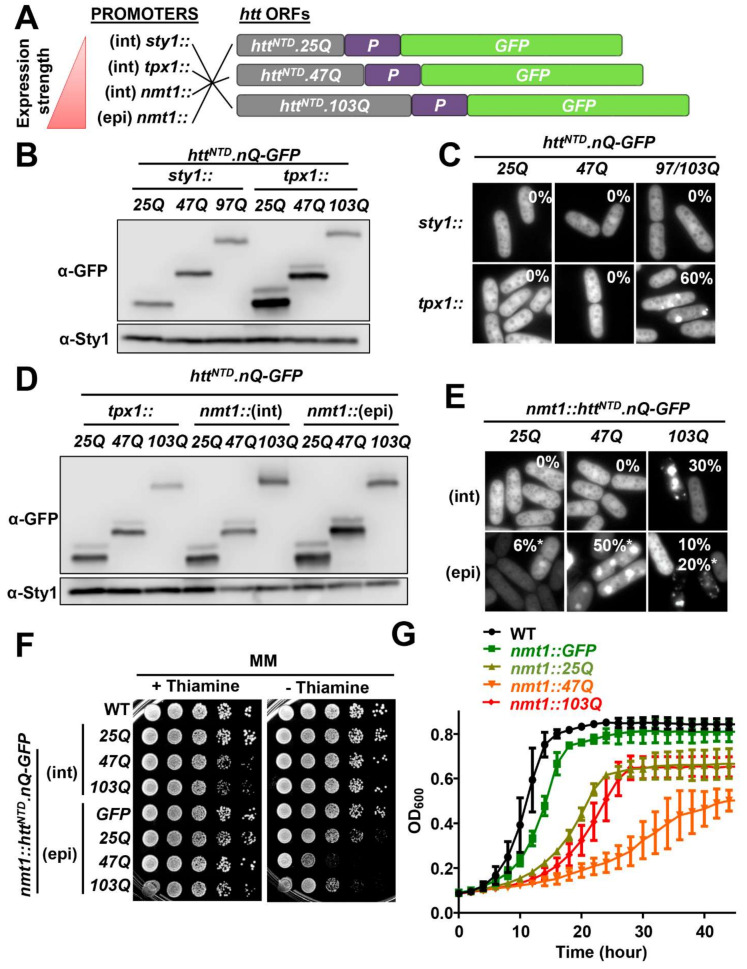
Characterization of the expression of Htt^NTD^.nQ-GFP in fission yeast. (**A**) Schematic representation of the Htt^NTD^.nQ-GFP constructs: the red triangle indicates the strength of the indicated promoters (int: integrative; epi: episomal); gray box: N-terminal region of human HTT containing a stretch of 25, 47 or 103 glutamines; purple box: proline-rich domain; green box: GFP. (**B**) Steady-state levels of Htt^NTD^.nQ-GFP determined by Western blot of TCA extracts from MM cultures of strains expressing Htt^NTD^.nQ-GFP driven by the *sty1* promoter (SG293.25Q, LM30.47Q and LM30.103Q) or by the *tpx1* promoter (AB1.25Q, AB1.47Q and AB1.103Q). Sty1 was used as loading control. (**C**) Fluorescence microscopy of same strains as above. The percentage of cells with aggregates is indicated in each micrograph. N = 90 in *tpx1*-driven 97Q/103Q constructs. (**D**) Steady-state levels of Htt^NTD^.nQ-GFP determined by Western blot of TCA extracts from strains grown in MM expressing Htt^NTD^.nQ-GFP driven by the *tpx1* promoter (AB1.25Q, AB1.47Q and AB1.103Q), by the integrative *nmt1* promoter (LM208.25Q, LM208.47Q and LM208.103Q) and by the episomal *nmt1* promoter (HM123 transformed with p659.25Q.3x, p659.47Q.3x and p659.103Q.3x). Sty1 was used as loading control. (**E**) Fluorescence microscopy of the strains expressing Htt^NTD^.nQ-GFP under the control of integrative and episomal *nmt1* promoters as in (**D**). The percentage of cells with aggregates is indicated in each micrograph. (*) indicates protein cumulus. N = 53 for integrative 103Q construct; N = 50, 54 and 60 for episomal 25Q, 47Q and 103Q constructs, respectively. (**F**) WT (972) and HM123 transformed with p690.3x (expressing GFP under the control of the *nmt1* promoter) and strains as in (**E**) were grown in MM and serially diluted and spotted on MM solid plates containing (repressed *nmt1*) or lacking (induced *nmt1*) thiamine. (**G**) Growth curves of MM cultures of wild-type (972) and strains expressing GFP or Htt^NTD^.nQ-GFP under the control of the *nmt1* promoter (HM123 transformed with p690.3x or with p659.25Q.3x, p659.47Q.3x and p659.103Q.3x respectively). Each curve represents the average of three biological replicates. Error bars represent SD.

**Figure 2 ijms-23-03950-f002:**
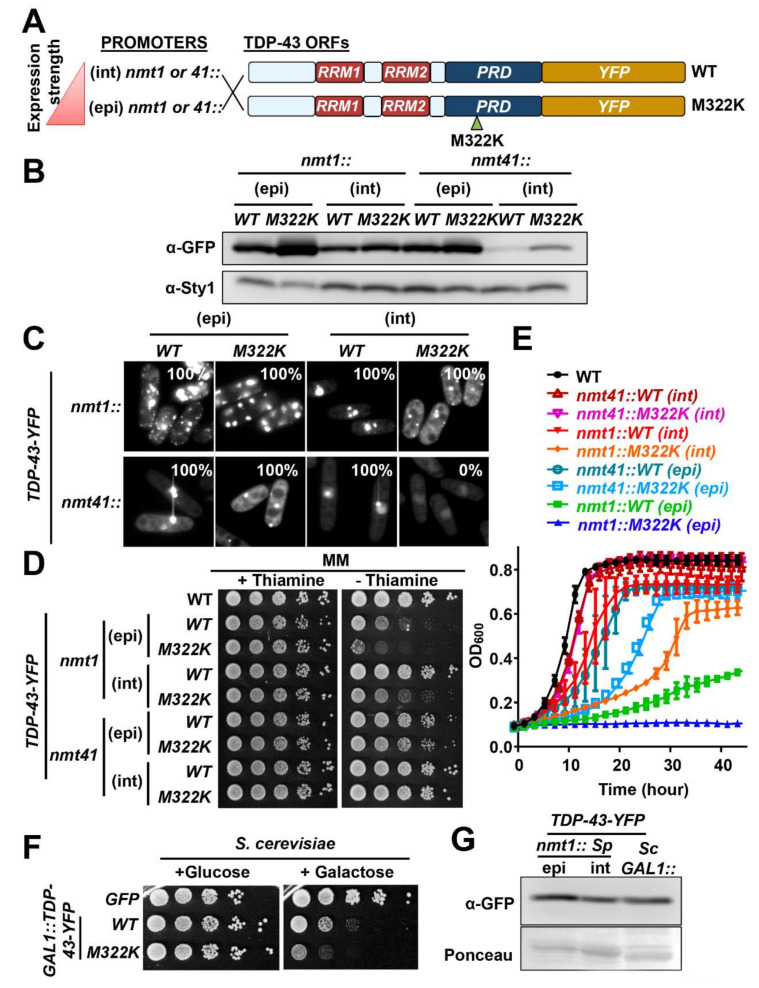
Characterization of the expression of TDP-43-YFP in fission yeast. (**A**) Schematic representation of the TDP-43-YFP constructs. The red triangle indicates the strength of the indicated promoters; red boxes: RNA recognition motive 1 and 2 (RRM1-2); blue box: prion-like domain (PRD); yellow box: YFP; small green triangle indicates the position of the mutation M322K within PRD. (**B**) Steady-state levels of TDP-43-YFP determined by Western blot of TCA extracts from MM cultures of strains expressing TDP-43-YFP and TDP-43.M322K-YFP under the control of the episomal *nmt1* promoter [HM123 transformed with p660.3x (*WT*) and p660.M322K.3x (*M322K*)], of the integrative *nmt1* promoter [LM218 (*WT*) and LM218.M322K (*M322K*)], of the episomal *nmt41* promoter [HM123 transformed with p660.41x (*WT*) and p660.M322K.41x (*M322K*)], and from the control of the integrative *nmt41* promoter [(LM233 (*WT*) and LM233.M322K (*M322K*)]. Sty1 was used as loading control. (**C**) Fluorescence microscopy of strains expressing TDP-43-YFP as in B. The percentage of cells with aggregates is indicated in each micrograph. N = 50 for all constructs. (**D**) WT (972) and strains used in (**C**) were serially diluted and spotted on MM solid plates as in Figure 1F. (**E**) Growth curves of MM cultures of wild-type (972) and cells expressing episomal *nmt1*-driven TDP-43-YFP and TDP-43.M322K-YFP [HM123 transformed with p660.3x (*WT*) and p660.M322K.3x (*M322K*)], integrative *nmt1*-driven TDP-43-YFP and TDP-43.M322K-YFP (LM218 and LM218.M322K), episomal *nmt41*-driven TDP-43-YFP and TDP-43.M322K-YFP [(HM123 transformed with p660.41x *(WT)* and p660.M322K.41x (*M322K*)] and integrative *nmt41*-driven TDP-43-YFP and TDP-43.M322K-YFP (LM233 and LM233.M322K) growing in MM. Each curve represents the average of three biological replicates. Error bars represent SD. (**F**) Strains expressing GFP, TDP-43-YFP or TDP-43.M322K-YFP under the control of the *GAL1* promoter (BY4741 transformed with pRS426-GFP, pRS416-TDP-43-YFP (*WT*) or pRS416-TDP-43.M322K-YFP (*M322K*) were serially diluted and spotted on synthetic complete media plates lacking uracil containing glucose (repressed *GAL1*) or galactose (induced GAL1). (**G**) Steady-state levels of TDP-43-YFP determined by Western blot of native extracts from *S. pombe* strains expressing integrative *nmt1*-driven TDP43-YFP (LM218) and from *S. cerevisiae* used strains expressing TDP-43-YFP under the control of the *GAL1* promoter. Ponceau staining was used as loading control.

**Figure 3 ijms-23-03950-f003:**
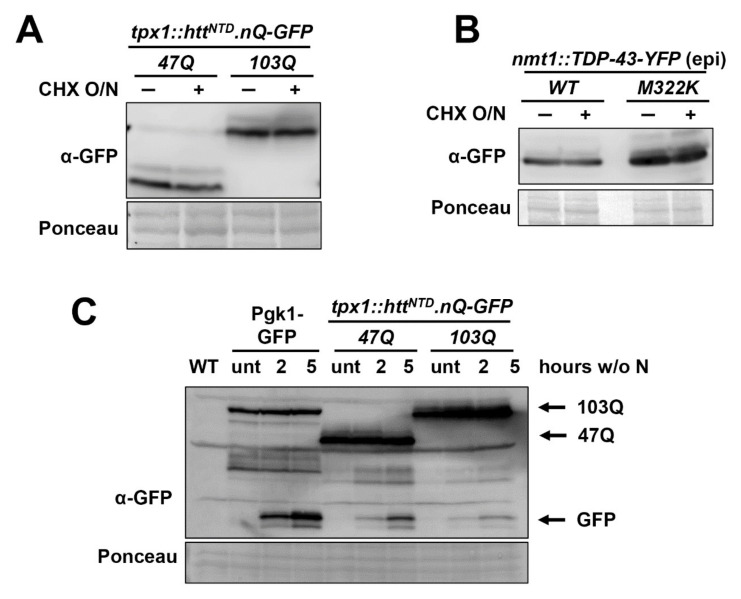
Htt^NTD^.nQ-GFP and TDP-43-YFP are not degraded by the UPS or autophagy. (**A**) The levels of Htt^NTD^.nQ-GFP were analysed by Western blot of TCA extracts from strains expressing *tpx1*-driven-Htt^NTD^.nQ-GFP (AB1.47Q and AB1.103Q) that were either untreated (−) or treated (+) overnight with 0.1 mg/mL cycloheximide (CHX). Ponceau staining was used as loading control. (**B**) The levels of TDP-43-YFP were analysed by Western blot of TCA extracts from strains expressing TDP-43-YFP or TDP43.M233K-YFP from an *nmt1* episomal plasmid [HM123 transformed with p660.3x (*WT*) and p660.M322K.3x (*M322K*)] that were grown in MM and treated as in A. Ponceau staining was used as loading control. (**C**) Levels of Pgk1-GFP, Htt^NTD^.nQ-GFP and cleaved GFP were determined by Western blot from TCA extracts obtained from a wild-type (972) and strains expressing Pgk1-GFP (RB44) or *tpx1*-driven Htt^NTD^.nQ-GFP (AB1.47Q and AB1.103Q) growing in the absence of a nitrogen source for the indicated time. Ponceau staining was used as loading control.

**Figure 4 ijms-23-03950-f004:**
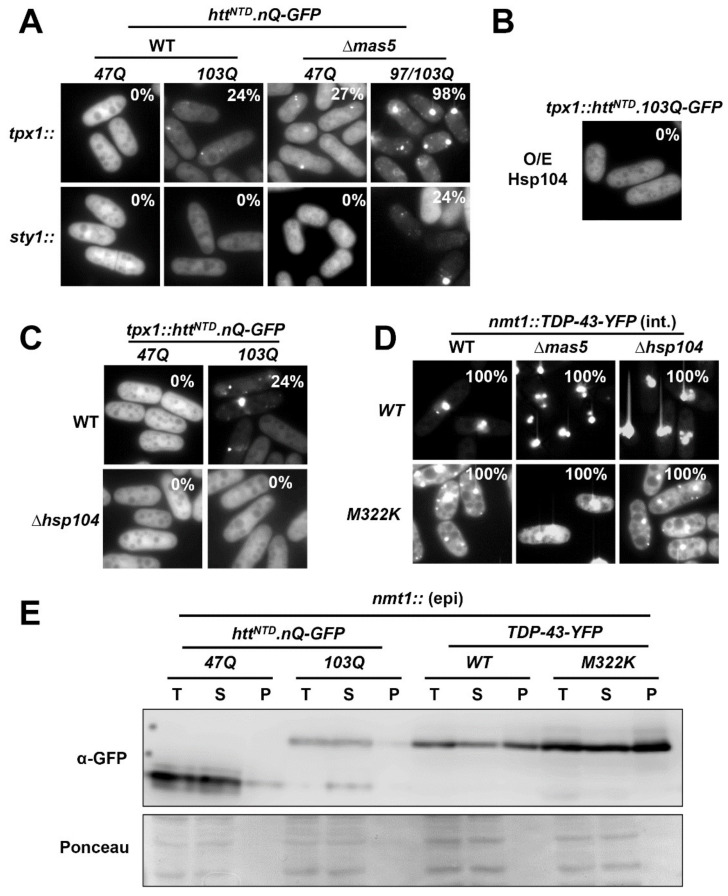
Components of the PQC system are involved in Htt^NTD^.nQ-GFP, but not TDP-43-YFP, aggregation. (**A**) Fluorescence microscopy of wild-type and ∆*mas5* strains expressing Htt^NTD^.nQ-GFP under the control of constitutive *tpx1* (WT: AB1.47Q and AB1.103Q; ∆*mas5*: SB413.47Q and SB413.103Q) or *sty1* promoter (WT: LM30.47Q and LM30.103Q; ∆*mas5*: LM18.47Q and LM18.103Q). The percentage of cells with aggregates is indicated. N = 55 for *tpx1*-driven 103Q constructs in WT cells; N = 100 and 80 for *tpx1*-driven 47Q and 103Q constructs in ∆*mas5* cells; N = 70 in *sty1*-driven 97/103Q in ∆*mas5* cells. (**B**) Fluorescence microscopy of cells expressing *tpx1*-driven HttNTD.103Q-GFP (AB6.103Q) transformed with p520 to over-express Hsp104 under the control of the episomal *nmt41* promoter. (**C**) Fluorescence microscopy of wild-type and ∆*hsp104* strains expressing Htt^NTD^.nQ-GFP under the control of constitutive *tpx1* promoter (WT: AB1.47Q and AB1.103Q; ∆*hsp104*: AB2.47Q and AB2.103Q). The percentage of cells with aggregates is indicated. N = 52 for 103Q construct in WT cells. (**D**) Fluorescence microscopy of wild-type, ∆*mas5* and ∆*hsp104* strains expressing TDP-43-YFP under the control of the conditional integrative *nmt1* promoter [WT: LM218 (WT), LM218.M322K (*M322K*); ∆*mas5*: LM225 (*WT*) and LM225.M322K (*M322K*); ∆*hsp104*: LM219 (WT) and LM219.M322K (*M322K*)]. The percentage of cells with aggregates is indicated. N = 50 for all constructs. (**E**) Solubility assay of Htt^NTD^.nQ-GFP and TDP-43-YFP in extracts from strains expressing Htt^NTD^.nQ-GFP or TDP-43-YFP under the episomal *nmt1* promoter [HM123 transformed with p659.47Q.3x (*47Q*) and p659.103Q.3x (*103Q*) or with p660.3x (*WT*) and p660.M322K.3x (M322K)] grown in MM. Protein levels in total (T), soluble (S) and pellet (P) fractions were determined by Western blot. Ponceau was used as loading control.

**Figure 5 ijms-23-03950-f005:**
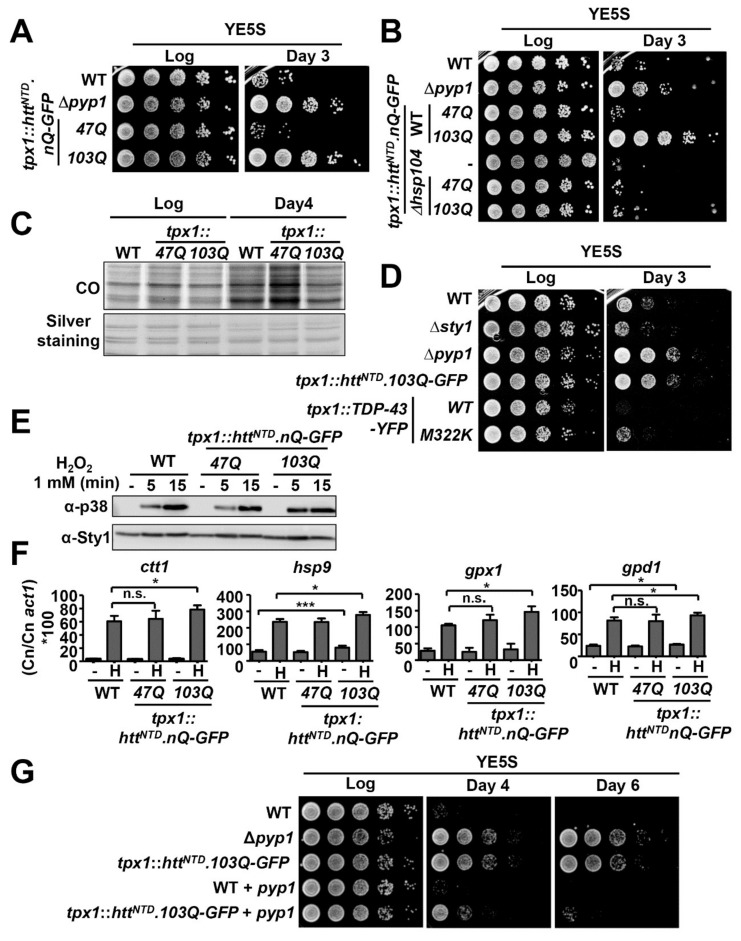
Expression of 103Q under the control of the constitutive *tpx1* promoter promotes lifespan extension. (**A**) Wild-type (972), ∆*pyp1* (EP48) and strains expressing *tpx1*-driven Htt^NTD^.nQ-GFP (AB1.4Q and AB1.103Q) were grown in YE5S for 3 days. Cultures from logarithmic (Log) or stationary phase (Day 3) were serially diluted and spotted onto YE5S plates. (**B**) Wild-type (972), ∆*pyp1* (EP48), strains expressing *tpx1*-driven Htt^NTD^.nQ-GFP (AB1.47Q and AB1.103Q), ∆*hsp104* (SG287) and ∆*hsp104* strains expressing *tpx1*-driven Htt^NTD^.nQ-GFP (AB2.47Q and AB2.103Q) were grown in YE5S and treated as in (**A**). (**C**) Protein carbonyl (CO) determination of extracts from wild-type (972) and strains expressing *tpx1*-driven Htt^NTD^.nQ-GFP (AB1.47Q and AB1.103Q) in logarithmic phase (Log) or in stationary phase (Day 4). Silver staining was used as loading control. (**D**) Wild-type (972), ∆*sty1* (AV18), ∆*pyp1* (EP48), strains expressing *tpx1*-driven Htt^NTD^.nQ-GFP (AB1.47Q and AB1.103Q) and strains expressing *tpx1*-driven TDP-43-YFP and TDP-43.M322K-YFP (LM210 and LM210.M322K) were grown in YE5S and treated as in (**A**). (**E**) Sty1 activation was analysed by Western blot with antibody α-p38 in TCA extracts from wild-type (972) and strains expressing *tpx1*-driven Htt^NTD^.nQ-GFP (AB1.47Q and AB1.103Q) that were either untreated (−) or treated with 1 mM H_2_O_2_ during 5 or 15 min. Sty1 was used as loading control. (**F**) The mRNA levels of Sty1-dependent (*ctt1*, *hsp9*, *gpx1*, and *gpd1*) genes from wild-type (972) and from strains expressing *tpx1*-driven Htt^NTD^.nQ-GFP (AB1.47Q and AB1.103Q) that were either untreated (−) or treated with 1 mM H_2_O_2_ for 15 min (H) were determined by RT-qPCR. Data are expressed as the mRNA copy number (Cn) relative to actin Cn and represent the average of at least three biological replicates. Error bars represent SD. Statistical significance was calculated between the indicated samples with an unpaired Student’s *t* test and 95% confidence level with *p* values of 0.05 (*) and 0.001 (***); ns, non-significant. (**G**) Wild-type (972), ∆*pyp1* (EP48), cells expressing Htt^NTD^.103Q-GFP (AB1.103Q), wild-type strains expressing an extra copy of *pyp1* under the control of its own promoter (SB706) and cells expressing Htt^NTD^.103Q-GFP with an extra copy of *pyp1* under the control of its own promoter (SB707) were grown in YE5S for up to 6 days and processed as in (**A**).

**Figure 6 ijms-23-03950-f006:**
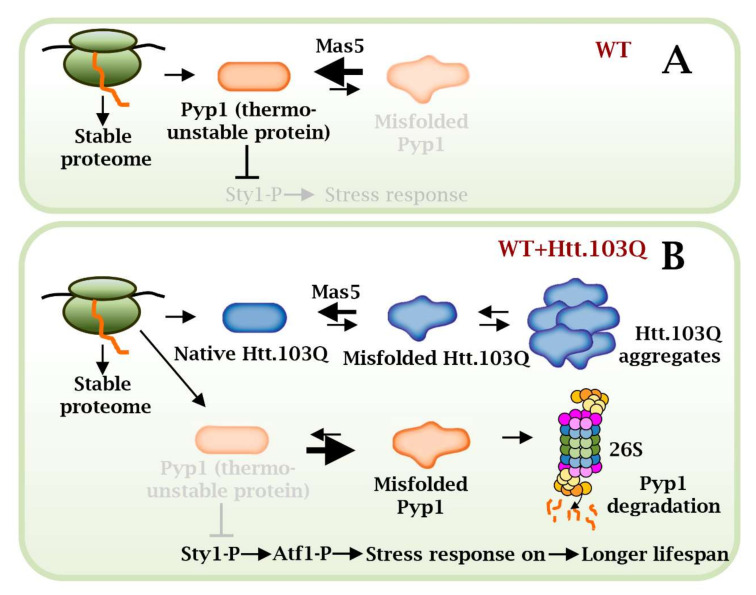
Scheme depicting a model of Mas5 role in in cells expressing or not Htt.103Q. (**A**) In basal conditions, Mas5 maintains the native conformation of the unstable Pyp1 phosphatase, which inhibits the MAP kinase Sty1. (**B**) However, in cells expressing Htt^NTD^.103Q-GFP, Mas5 participates in Htt^NTD^.103Q-GFP folding and leaves Pyp1 partially unattended. This promotes misfolding and degradation of Pyp1, with the subsequent activation of the Sty1- and Atf1-dependent stress response.

**Table 1 ijms-23-03950-t001:** Intracellular concentrations of Htt and TDP-43 derivatives expressed in *S. pombe*.

Promoter/Type of Plasmid ^a^	Intracellular Concentration of Htt.nQ-GFP (µM)	Intracellular Concentration of TDP-43-YFP (µM)
*sty1*/integrative	1	N.D.
*tpx1*/integrative	3	3
*nmt41*/integrative	N.D.	0.8 (WT), 1.6 (M322K)
*nmt41*/episomal	N.D.	5
*nmt1*/integrative	4	4
*nmt1*/episomal	9	5 (WT), 9 (M322K)

^a^ The *sty1* and *tpx1* are constitutive promoters, whereas *nmt1* is expressed upon thiamine withdrawal. While integrative plasmids have only one copy per cell, episomal plasmids have on average 7–8 copies per cell [43].

## Data Availability

The data that support the findings of this study are available from the corresponding authors upon reasonable request.

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
