# Peer review of "Expression of Huntingtin and TDP-43 Derivatives in Fission Yeast Can Cause Both Beneficial and Toxic Effects"

_ijms, 2022, doi:10.3390/ijms23073950_

Round 1

Reviewer 1 Report

In this work, Marte et al. have compared the effects of different TDP-43 and Huntingtin proteins aggregation in fission yeast (as opposed to many previous studies done on these proteins which have been performed in budding yeast). Considering that the aggregation of these proteins is a major hallmark of several neurodegenerative diseases, a side-by-side comparison of their activities is a topic that is both timely and generally interesting from the point of view of establishing connections and highlighting differences between these disease proteins. From a technical point of view, the authors have followed standard expression strategies in yeast and the experimental pipeline seems very robust. With one exception, there are only a few minor concerns that should be addressed by the authors:

  • Figure 1G should contain error bars and statistical variations (since they represent the average of three biological replicates)
  • A growth curve graph similar to Fig.1G should also be provided in Figure 2 for the TDP-43 M322K mutant.
  • At line 128 in the introduction the “were unlikely affected” should be replaced with “were differentially affected”.
  • At line 391 (Figure 4BC) should be Figure 4B-C)

Author Response

Reviewer 1

In this work, Marte et al. have compared the effects of different TDP-43 and Huntingtin proteins aggregation in fission yeast (as opposed to many previous studies done on these proteins which have been performed in budding yeast). Considering that the aggregation of these proteins is a major hallmark of several neurodegenerative diseases, a side-by-side comparison of their activities is a topic that is both timely and generally interesting from the point of view of establishing connections and highlighting differences between these disease proteins. From a technical point of view, the authors have followed standard expression strategies in yeast and the experimental pipeline seems very robust. With one exception, there are only a few minor concerns that should be addressed by the authors:

We are very thankful to the reviewer for her/his fair and careful consideration of our manuscript, for the time invested in the revision process, and for the kind words about our work.  We have introduced all the changes suggested.  All these changes have been indicated in the manuscript file.

  • Figure 1G should contain error bars and statistical variations (since they represent the average of three biological replicates)
  •  

Thanks for the comment.  Error bars were included in Fig. S1G.  But have moved it to Fig. 1G.

  •  
  • A growth curve graph similar to Fig.1G should also be provided in Figure 2 for the TDP-43 M322K mutant.

The corresponding growth curves for TDP-43 were already shown as Fig. S2B.  We have now moved it to Fig. 2E.

  • At line 128 in the introduction the “were unlikely affected” should be replaced with “were differentially affected”.

Our experiments suggest that autophagy and UPS are not affected by expression of Htt or TDP-43 derivatives.  We have replaced “were unlikely affected” with “were not affected”.

  • At line 391 (Figure 4BC) should be Figure 4B-C)

Changed, thanks!

Reviewer 2 Report

In this manuscript, Marte and colleagues study the effects of expressing Huntingtin and TDP-43 derivative proteins in the model fission yeast. In particular, the authors use various promoters to express Huntingtin proteins bearing different lengths of polyglutamine arrays and assess the aggregation of such proteins and their effects on fitness. Interestingly, the authors conclude that these proteins are toxic when expressed at extremely high levels but they can have beneficial effects at moderate concentrations. Overall, the work is well-executed as the experimental data is of good quality. Please find my comments below.

Major comments:

_ Fig. 6: The authors come up with a model in which Sty1 is activated upon expression of Htt-103Q and that would trigger a longer lifespan. Therefore, an obvious control to be added in the experimental data is a sty1 mutant, ideally a non-activable T171/T173 phospho-mutant, confirming that indeed the longer lifespan depends on Sty1 activation.

_ Regarding the effects on protein aggregation, there are so many constructs expressed in various conditions and mutant backgrounds that it makes it difficult for the reader to have the overall picture.  Please add a supp. Table summarizing such data.

Minor issues:

_ Western blots (e.g. Fig. 1B-D, 3A, S2A, S3B, S5A): It is noticeable that Htt proteins, mostly the 25Q and 47Q constructs, display a second upper band. I wonder if this is the result of amplification of the repeats encoding polyglutamine tracts. Could the authors comment on that?

_ The number of cells analyzed (i.e. N=) must be indicated in figure legends.

_ Nomenclature used in some figures is awkward as follows:

  • 5G and S5D: “pyp1-pyp1” or “pyp1::pyp1”. Is that a transgene? It is not obvious what it means.
  • S3C: “WT WT”. It may be better to indicate two sets, one with TDP-43 and other with TDP-43M322K.

_ Language can be improved as follows.

  • Line 22: “simple model system” sounds pejorative. “Highly tractable unicellular model…” or something like that would be better.
  • Line 144: “against them”. Not clear what it means.
  • Line 230: “10-times” must read “10-fold”

_ Typos:

  • Line 50: “believe” must read “belief”
  • Line 304: “later” must read “latter”
  • Line 619: “uracile” must read “uracil”

Author Response

Reviewer 2

In this manuscript, Marte and colleagues study the effects of expressing Huntingtin and TDP-43 derivative proteins in the model fission yeast. In particular, the authors use various promoters to express Huntingtin proteins bearing different lengths of polyglutamine arrays and assess the aggregation of such proteins and their effects on fitness. Interestingly, the authors conclude that these proteins are toxic when expressed at extremely high levels but they can have beneficial effects at moderate concentrations. Overall, the work is well-executed as the experimental data is of good quality. Please find my comments below.

We are very thankful to the reviewer for her/his fair and careful consideration of our manuscript, for the time invested in the revision process, and for the kind words about our work.  We have introduced all the changes suggested.  All these changes have been indicated in the manuscript file.

Major comments:

_ Fig. 6: The authors come up with a model in which Sty1 is activated upon expression of Htt-103Q and that would trigger a longer lifespan. Therefore, an obvious control to be added in the experimental data is a sty1 mutant, ideally a non-activable T171/T173 phospho-mutant, confirming that indeed the longer lifespan depends on Sty1 activation.

Cells lacking Sty1 display pleiotropic defects (the same occurs with non-activatable mutant), and it is not surprising that, when combined with any given mutation conferring beneficial traits, sty1 deletion suppresses them.  Instead, we combined the expression of 103Q with deletion of the transcription factor-coding gene, atf1, the main transcriptional effector of Sty1: this deletion fully suppresses the longevity associated to 103Q.  This is provided in Fig. S5C, and it is explained in lines 488-490.  To ease the interpretation of the data, we have added Atf1 in the scheme of Fig. 6, to clearly show that the effects of Sty1 activation rely on the activation of the downstream transcription factor; this has also been added to the Discussion (line 557).

_ Regarding the effects on protein aggregation, there are so many constructs expressed in various conditions and mutant backgrounds that it makes it difficult for the reader to have the overall picture.  Please add a supp. Table summarizing such data.

We have now generated a new Table S3 summarizing the % of aggregation in all the different backgrounds.

Minor issues:

_ Western blots (e.g. Fig. 1B-D, 3A, S2A, S3B, S5A): It is noticeable that Htt proteins, mostly the 25Q and 47Q constructs, display a second upper band. I wonder if this is the result of amplification of the repeats encoding polyglutamine tracts. Could the authors comment on that?

Thanks for pointing this issue.  It is very clear in Western blots that 25Q and 47Q constructs show a slower migrating band, however we do not think this is specific for these two constructs as it can also be observed, to a lower extent, in the 103Q construct.  We think that the only reason for being more clear in 25Q and 47Q is the larger amount of 25Q-GFP and 47Q-GFP compared to 103Q-GFP. 

We did observe polyQ contraction in 103Q and 72Q constructs in E. coli during the cloning process before transformation in S. pombe, shifting to 97Q and 68Q, respectively, but, although we have not tested the polyQ length stability in our S. pombe strains, we do not think the retarded bands correspond to polyQ expansions.  First, it has already been published by Supattapone’s group that polyQ constructs are genetically more stable in S. pombe than in S. cerevisiae, and to the best of our knowledge, there are no different genetic variants in S. pombe that could explain different polyQ stabilities between different labs.  Second, the distance between the more prominent band and the retarded upper band is the same for all three constructs.  Were these different bands arising from expansions, this would mean that the increase for the 103Q construct would be much larger than the increase observed in 25Q, as the resolution in SDS-PAGE is not linear.  While, this is still formally possible and we have not discarded it, we believe instead that the different mobility bands are probably due to a delay in the electrophoresis either during sample entrance into the stacking gel or during the stacking separation.

_ The number of cells analyzed (i.e. N=) must be indicated in figure legends.

We have now added in the figure legend the number of cells analyzed for all strains where Htt-GFP formed aggregates.

_ Nomenclature used in some figures is awkward as follows:

  • 5G and S5D: “pyp1-pyp1” or “pyp1::pyp1”. Is that a transgene? It is not obvious what it means.

We have changed the nomenclature in figures 5G and S5D to make it less confusing.  We think that now it is more clear that the indicated strains bear an extra copy of the pyp1 gene, as it is also explained in the corresponding figure legends.

  • S3C: “WT WT”. It may be better to indicate two sets, one with TDP-43 and other with TDP-43M322K.

Sure, done.  All the Htt and TDP-43 derivatives are now separated (Fig. S3C)

_ Language can be improved as follows.

  • Line 22: “simple model system” sounds pejorative. “Highly tractable unicellular model…” or something like that would be better.

Done, thanks

  • Line 144: “against them”. Not clear what it means.

We have changed it: specific polyclonal antibodies

  • Line 230: “10-times” must read “10-fold”

Done, thanks

_ Typos:

  • Line 50: “believe” must read “belief”

Thanks, changed

  • Line 304: “later” must read “latter”

Done, thanks

  • Line 619: “uracile” must read “uracil”

We have changed it in 5 different positions, thanks a lot

Round 2

Reviewer 2 Report

I am satisfied with the authors' response. The ms. is now suitable for publication.